# Posture and Health: Are the Biomechanical Postural Evaluation and the Postural Evaluation Questionnaire Comparable to and Predictive of the Digitized Biometrics Examination?

**DOI:** 10.3390/ijerph18073507

**Published:** 2021-03-28

**Authors:** Giovanni Barassi, Edoardo Di Simone, Piero Galasso, Salvatore Cristiani, Marco Supplizi, Leonidas Kontochristos, Simona Colarusso, Christian Pasquale Visciano, Pietro Marano, Di Iulio Antonella, Orazio Giancola

**Affiliations:** 1Physiotherapy, Rehabilitation and Reeducation Training Center (CeFiRR), Venue Gabriele d’Annunzio, School of Medicine and Health Sciences, University of Chieti-Pescara, 66100 Chieti, Italy; edoardo.disimone@gmail.com (E.D.S.); msupplizi@gmail.com (M.S.); lekontochristos@gmail.com (L.K.); visciano77@gmail.com (C.P.V.); 2San Raffaele University, 00166 Rome, Italy; pierogalassomail@gmail.com; 3Local Health Company RM2, 00159 Rome, Italy; cristianisalvatore@gmail.com; 4Communication, Research, Innovation, Department of Communication and Social Research, Sapienza University, 00185 Rome, Italy; simona.colarusso@gmail.com; 5Department of Neuro-Rehabilitation, Madonna del Rosario Clinic, 95125 Catania, Italy; pietromarano.pm@gmail.com; 6U.O.C. Thoracic Surgery, Santo Spirito Hospital Pescara, ASL Pescara, 65124 Pescara, Italy; antonella.diiulio01@gmail.com; 7Department of Social and Economic Sciences, Sapienza University, 00185 Rome, Italy; orazio.giancola@uniroma1.it

**Keywords:** postural evaluation, forward head posture, myofascial disorders

## Abstract

Background: Postural tone alterations are expressions of myofascial and, therefore, of structural, visceral, and emotional disorders. To prevent these disorders, this study proposes a quantitative investigation method which administers a postural evaluation questionnaire and a postural biomechanical evaluation to 100 healthy subjects. Methods: The reliability of the method is studied by comparing both assessments with digitized biometrics. In addition, 50 subjects undergo the biomechanical evaluation form twice, by four different operators, to study the intraoperative repeatability. Results: The results show a satisfactory overlap between the results obtained with the postural evaluation questionnaire and the postural biomechanical evaluation compared to computerized biometrics. Furthermore, intraoperative repeatability in the use of the biomechanical evaluation form is demonstrated thanks to a minimal margin of error. Conclusions: This experience suggests the importance of undertaking this path in both the curative and the preventive sphere on a large scale and on different types of people who easily, and even unknowingly, may face dysfunctional syndromes, not only structural and myofascial but also consequently of the entire body’s homeostasis.

## 1. Introduction

Posture can be defined as the position of joints and parts of the body in an upright, sitting, or lying position. Indeed, the adult’s vertebral column is composed of two main and two compensatory curves. In a baby, the main curves are in the thoracic and sacral regions and have posterior convexity. The compensatory curves are developed in the standing position in the cervical and lumbar regions so have anterior convexity [1,2]. The shape of the entire body depends on these functional stimulations [3,4]. These curves continue to develop in the child until the onset of growth spurts. In fact, at the age of three weeks, a structure forms in the three-leaf organism with the mesoderm interposed where the ectoderm and the endoderm were, comparable to an “epithelium”. If we observe the body in terms of its functional “development”, a picture emerges where each structure has its function and, subsequently, its shape [5]. In all, the muscle chain and fascial continuity appear to be an evolutionary expression of the spinal cord that develops in the lateral chain for needs dictated by the ectoderm that develops into the central nervous system [6]. Therefore, it is possible to define posture as the “spinal cord reflex manifestation of convergences and facilitation, secondary to cortical expressiveness”.

These body systems refer to the evolution of the paraxial mesodermal germination muscle system (skeletal musculature derived from the somites extending from the occipital to the sacral region and from somitomers to the head region) or splanchnic mesodermal derivation (smooth muscle), which control the training and operating scheme. Since the spinal cord housed in the vertebral column supports the head, once the secondary curves are formed, head positioning, the relationship of mandible with maxilla, and the relationship of mandibular teeth with maxillary teeth all play a role in the stabilization of the cervical spine. For the spine to remain neutral, core muscles and central nervous system functionality play the role of stabilizers. In normal conditions, spinal curves are able to adapt themselves to gravity and external forces. To be in equilibrium, spinal curves should be in balance. Conversely, the continued stabilization of the spine’s muscles and joints is needed to achieve balance in an incorrect posture. Some studies demonstrated that postural alterations could change the mechanical sensitivity of different tissues, decreasing their tolerance to mechanical stress [7]. Postural alterations can affect head position due to its adaptation to the movements. It is common to record a forward head posture (FHP) in symptomatic humans [8]. Study of asymptomatic individuals has proven that FHP can increase the mechano-sensitivity of some cervical tissues and moderate the relationship between FHP and neck pain (NP) in the vertebral column. Moreover, individuals with FHP and subclinical NP showed a release after the application of pressure in the upper trapezius [9]. FHP could be linked to temporomandibular joint positions. Additionally, the connection between the spine and the temporomandibular joint is crucial for the positioning of the column within the space. The temporomandibular joint affects physiological curves and posture. In the body, incorrect posture could cause muscle imbalance in terms of strength and elasticity. Indeed, some muscles appear contracted, while others appear stretched. Both conditions cause stiffness and weakness. Spinal imbalance in improper posture leads to mechanical distress of the muscles, joints, and tissues [10]. This observational study compares the measurement of the Postural Evaluation Questionnaire (PEQ) and Biomechanical Postural Evaluation (BPE) with Digitized Biometrics Examination (DBE). 

The aim of this study is to see if the outcome of a less expensive and more quickly administrable postural evaluation, such as PEQ and BPE, is comparable to, and as predictive as, the result of a more sophisticated and reliable, but also more expensive, postural assessment, such as DBE. DBE was considered in this study, but there are other tools that objectively evaluate static balance, such as triaxial accelerometers [11,12]. A less expensive and quickly administrable postural evaluation could be administered in a preventive setting on a large scale in order to prevent dysfunctional syndromes. This would have a positive impact on social costs and work activity.

## 2. Materials and Methods

This observational study was conducted in 2019 at the Physiotherapy, Rehabilitation, and Reeducation Center (CeFiRR), Training Centre venue “Gabriele d’Annunzio” University of Chieti-Pescara, School of Medicine and Health Sciences, located at Viale Abruzzo 322, Chieti Scalo. All participants signed the informed consent for the evaluation procedure, which complies with the latest revision of the Helsinki Declaration and with the procedures defined by the ISO 9001-2015 standards for research and experimentation; this procedure also protects the privacy of subjects participating and defines the procedures in biomedical research. The sample consisted of 100 healthy students of d’Annunzio University, including 82 males and 38 females, aged between 20 and 43 years, with an average age of 23.38 years. 

The PEQ is a set of 63 questions that investigate a person’s health status quickly and comprehensively. It investigates conditions or behaviors that relate to postural disorders. It was completed by the patients themselves before the BPE. Subsequently, the questionnaire was analyzed by the examiner in the presence of the subject, to further investigate the condition of the latter and to clarify any misunderstandings.

The Questionnaire consisted of two parts:the first collected personal, family, and employment datathe second investigated the clinical conditions, the biomechanical structure, and the receptor structure

The questionnaire’s score was the sum of the individual Postural Biomechanical Index (PBI), which is directly proportional to the deviation from the norm of the investigated parameters. Therefore, a high score meant a greater probable deviation from the norm, while a relatively low score meant a probable condition of normality, or less deviation from it.

The BPE is a manual analysis of the osteo-myofascial system, aimed at determining the subject’s overall postural condition. It consists of the application of pressure stimuli, of about 2 kg, in specific points and the consequent quantification of the pain evoked by the subject (Table 1).

To define the intensity of pain evoked by each stimulus, the 100 subjects were given the NRS (Numerical Rating Scale). The NRS allows pain to be rated from 0 to 10, where 0 corresponds to an absence of pain and 10 to unbearable pain. The NRS was verbally administered by asking all patients the same question: “On a scale from 0 to 10, where 0 represents the absence of pain and 10 the worst possible pain, what is your level of pain?”. To study its reproducibility, the BPE was repeated once on a sample of 50 subjects by four different operators and after 6 days.

The DBE, provided by Diasu Health Technologies, Rome, Italy is a non-invasive investigation method developed for the quantitative and qualitative evaluation of a subject’s equilibrium strategy. The sample was subjected to static pressure test and stabilometric examination to analyze the percentage difference in load and the Romberg Index (RI) of each subject.

The difference in load percentage allows us to observe if the load is homogeneously distributed between the right and left limbs. Variations of more than 3% are considered an index of dysfunction. Measuring body weight distributions allows us to evaluate probable postural alterations [13].

The stabilometric examination studies the body’s oscillations in an orthostatic position. Measuring body swings allows us to evaluate the postural receptors’ performance. The oscillations of body structure in maintaining balance in the orthostatic position are measured by analyzing the displacement of the Center of Pressure (CoP). The stabilometric data were acquired on a force platform under standardized Association Française de Posturologie (AFP) conditions: subject barefoot, feet oriented at 30°, heels 2 cm apart, arms alongside the body, visual target 90 cm away in front of the subject, and acquisition time of 51.2 s [14].

## 3. Results

The analysis of the data relating to the sample consisted of three phases. The first phase included descriptive analysis, the second phase included linear correlations between the variables considered, while the third phase included linear regression models.

For each model, the share of variance explained (R-square), the significance of the model (ANOVA value), and finally the coefficients (with relative significance) were presented.

The coefficient of variation (Table 2) is a dispersion index that makes it possible to compare measurements of phenomena referring to different units of measurement, as it is a dimensionless quantity (i.e., not referring to any unit of measurement). In the case of the scoring of the PEQ and of the BPE measurement, the coefficient showed normal and not excessively dispersed distributions. In the case of the Romberg Index, it increased significantly but remained within the parameters (since it did not exceed the value of one).

There was a strong and significant correlation between the scores obtained through the PEQ and the BPE (Table 3).

The most important aspect to point out is the strong and completely significant correlation between the score on the PEQ and the “load percentage difference” (+0.665). The BPE data sheet, which was in turn correlated to the score obtained through the PEQ, also correlated to the “load percentage difference” (+0.512), but to a lesser extent.

There was also a correlation above the significance threshold for *p* 0.05 between the Romberg Index and the measure obtained through the PEQ.

Having observed the correlation between the “PEQ score” and “load percentage difference”, we opted for an analysis using a scatter plot and linear regression estimation (Figure 1). To ensure correct representation, given the different units of measurement and variation fields of the PEQ score and the percentage difference in load, the two measures were subjected to standardization by producing z-scores. (It is important to underline that standardization facilitates the geometry reading of the relationship between variables but does not alter the correlation and regression values.)

The results allow us to state that the PEQ score is a reliable and adequate predictor of the load percentage difference.

To further corroborate this hypothesis, we developed three sequential regression models (Table 4). In all three models, the dependent variable is the “load percentage”. In the first model, we used the “PEQ score” as the regressor/predictor; in the second model, we used the BPE score, and finally, in the third and last model, we used both measures simultaneously (“PEQ score” and “BPE score”).

Consistent with the correlations, the “PEQ score” was shown to be an excellent regressor (Mod.1). The result of the BPE score also yields an acceptable result (Mod.2). Finally, in the third (Mod.3), we used both measurements simultaneously, observing the values of the coefficient Beta (which, being standardized, are comparable to each other). The “PEQ score” was shown to be a more accurate predictor of the “BPE score”, even if the combined use of the two measures led to a slight increase in the overall R-square (and with full statistical significance of the two coefficients).

Finally, studying the reproducibility of BPE, it emerged that:the BPE reproducibility error was 0.63;a minor reproducibility error was found during the evaluation of the right metatarsus, with a value of 0.14;the greatest reproducibility error was found in the evaluation of the right piriformis muscle, with a value of 1.7.

## 4. Discussions

The spinal cord’s functions include not only the structural support of the body, but it also contains neural elements while allowing proper interaction with the brain. Indeed, different types of tissue are represented on it. In the fetal period, the main curve of the vertebral column is kyphotic. However, the cervical and lumbar lordosis curves develop secondarily after birth. In particular, the lumbar lordosis develops as a result of the infant achieving a sitting, and then standing, posture [15]. The association between postural alterations and pain is still discussed with the role of posture in NP [16]. In patients with musculoskeletal pain disorders, posture analysis could be the first assessment for discovering and improving risk factors and finding a reasonable prevention strategy [17]. One of the common posture misalignments of the spine is head protrusion accompanied by extension of the upper cervical spine and flexion of the lower cervical spine. It is called forward head posture (FHP) [18]. Most likely, FHP is a response to increased upper thoracic kyphosis, with the aim of increasing the lower cervical neural foraminal area to alleviate nerve root compression and to reduce the burden on posterior muscles. Conversely, the hyperextension of the upper cervical segments (C0–C1–C2), necessary to keep the gaze horizontal, causes a reduction in the cervical neural foraminal areas [19]. FHP can be linked to a person’s lifestyle and profession, which were carefully investigated in the questionnaire. In particular, Kyung-soon et al. reported that head anteriorization increases with increasing duration of computer usage [20]. Furthermore, the high dynamic load at the level of C1–C2 may be responsible for the Y-shaped trabecular bone structure in the odontoid process [21]. The association between FHP and NP is uncertain and age may play an important role. In fact, some studies have shown that adults with NP have more FHP than asymptomatic participants [22,23], while other studies have shown that adolescents with NP have less FHP than asymptomatic participants [24]. Clinical observations suggest that kyphosis reduces cervical mobility through an increase in forward head posture (FHP) [25]. The questionnaire investigated the presence of headache and NP, because FHP and weakness of the upper cervical flexor muscles could be associated with chronic tension-type headaches (CTTH), and it could be difficult to differentiate the origin of the symptoms. Furthermore, it was crucial to determine the relationship between CTTH and poor cranio-cervical alignment. Thus, postural correction and re-education could be an integral part of both the prevention and management of patients with cervical headache. CTTH usually follows a protracted course. Indeed, 65% of patients surveyed by Jull had a history of headache ranging from 2 to 20 years [26]. Fernández-de-las-Peñas demonstrated the relationship between headache and FHP in patients with CTTH. Indeed, the greater the FHP, the lower the neck mobility. Moreover, in this population, there was a positive correlation between FHP and headache frequency and a negative correlation between neck mobility and headache parameters [27]. Other studies have analyzed posture and balance strategies, but the samples always comprise symptomatic individuals [28,29]. For instance, balance in patients with chronic or post-surgical low back pain has been analyzed. This study, instead, is based on pain evoked and reported in asymptomatic individuals, in order to develop a protocol for the prevention of dysfunctional syndromes. Clinical pain is not simply the expression of a particular stimulation, but it reflects the rate of central nociceptive circuits. When central sensitization occurs, nociceptor inputs could trigger a prolonged but reversible increase in the excitability and synaptic efficacy of neurons in central nociceptive pathways [30]. However, during inflammation, plasticity changes could occur in the peripheral and central nervous systems, which lower the pain thresholds. Consequently, they could give rise to allodynia (pain in response to a normally innocuous stimulus) and hyperalgesia (heightened pain intensity in response to a normally painful stimulus). Indeed, various musculoskeletal disorders, such as shoulder impingement syndrome, FHP, or epicondylalgia, are characterized by a lowering of pain thresholds in healthy tissues, and they can develop dysfunctional disorders [31]. In some pathologies typical of old age, such as arthritis, the source of pain may be the nerve degeneration typical of old age, rather than the disease itself [32].

The analysis of the results confirmed the comparability of the PEQ and BPE outcomes with the DPE outcomes. Therefore, this study introduces an innovative, non-invasive, fast-performing, economical, and repeatable method of investigation of posture, consisting of a questionnaire and a manual biomechanic analysis. This method analyzes posture by researching and contextualizing every alteration which may influence the postural tone. The information obtained by the PEQ, a summary of the osteo-myofascial, visceral, and emotional conditions of the subject, was shown to be a significant predictor of postural and body homeostasis conditions. Even the total score of the BPE was found to be a useful predictor, to a lesser extent. However, it is the combination of the two methods of survey which has the greatest reliability, revealing a slight increase in the total R-square with a confirmed statistic of the two coefficients. This study has limitations: the correlation between some body subsystems investigated by the questionnaire and posture has not yet been clarified. Comorbidities may have affected the results. Indeed, basic muscle tone is the expression of varied afferential information from visceral, emotional, and structural receptors. Each parameter at the spinal cord level is modulated by the autonomic nervous system (in particular, by the orthosympathetic system), which influences vascular function, internal organs, and basic muscle tone. Therefore, in the vertical functions of the spinal cord, there is a segmental (horizontal) manifestation, which expresses the dysfunction in the dermo-myotonic field. Therefore, myofascial and postural tonic dysfunction can be influenced by pre-existing pathologies or comorbidities, even asymptomatic ones [33].

Furthermore, only some values of the DBE were considered. Previously, Blondel, B. proposed a postural analysis protocol using a strength platform and skin markers [34]. However, the complexity of this protocol makes it difficult to repeat. This study, on the other hand, suggests a protocol based on a multidisciplinary approach, which can be carried out in any environmental situation, with minimal economic impact and with a result that can reflect the same values obtained by a DBE (R-square = 0.271). This method, aimed at asymptomatic subjects, could also be applied to symptomatic subjects and subjects of all ages. In fact, with increasing age, there is a static balance deterioration [35], which could cause an increase in the risk of falls. Targeted intervention, based on specific exercises aimed at recovering skills related to postural control, could reduce the risk of falls and hospitalizations. This method also provides the grounds for implementation in the preventive and curative field on a large scale and for different subjects who could easily experience dysfunctional syndromes, not only structural and myofascial but also related to whole-body homeostasis. Especially in this pandemic period, which has made people more sedentary, dysfunctional syndromes could be prevented and this would have a positive impact on social costs and work activity.

## Figures and Tables

**Figure 1 ijerph-18-03507-f001:**
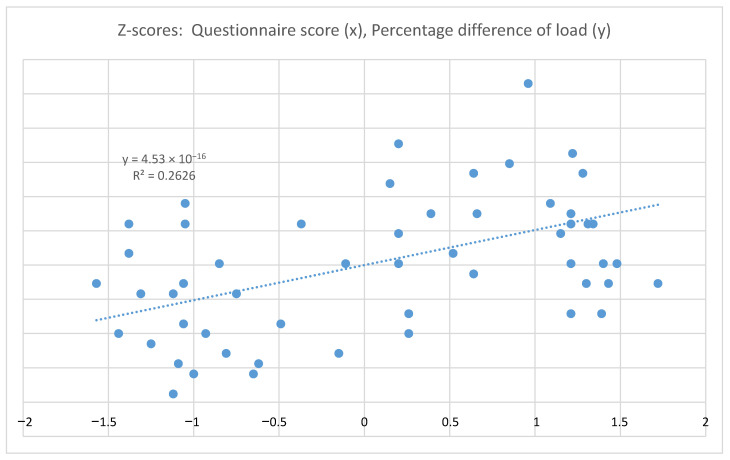
Scatter plot graphic (with linear interpolation and regression equation).

**Table 1 ijerph-18-03507-t001:** Structures examined by the Biomechanical Postural Evaluation (BPE), divided by body areas.

Foot	I–V Metatarsus	Midfoot	Calcaneus	Achilles Tendon	Anterior Talo-Fibular Ligament	Posterior Talo-Fibular Ligament	Calcaneo- Fibular Ligament	Deltoid Ligament
**Limb**	Gastrocnemius	Soleus	Peronei longi	Tibialis anterior				
**Thigh**	Tensor fascia lata	Harmstring	Long adductor					
**Pelvis**	Ileopsoas	Piriformis						
**Rachis**	Quadratus lumborum	Erector columnae	Trapezius	Pectoralis minor	SCOM	Masseter		

**Table 2 ijerph-18-03507-t002:** Coefficient of variation.

Assessments	Variation Coefficient
Romberg Index	0.71
Load percentage difference	0.55
PEQ Score	0.36
BPE Score	0.47

PEQ: Postural Evaluation Questionnaire; BPE: Biomechanical Postural Evaluation.

**Table 3 ijerph-18-03507-t003:** Correlation matrix.

	Measures	Romberg Index	Load Percentage Difference	PEQ Score	BPE Score
Romberg Index	Pearson correlation	1	0.070	0.220 *	0.152
	Sign. (two-tailed)		0.491	0.028	0.131
	*N*	100	100	100	100
Load percentage difference	Pearson correlation	0.070	1	0.512 **	0.412 **
	Sign. (two-tailed)	0.491		0.000	0.000
	*N*	100	100	100	100
PEQ score	Pearson correlation	0.220 *	0.512 **	1	0.665 **
	Sign. (two-tailed)	0.028	0.000		0.000
	*N*	100	100	100	100
BPE Score	Pearson correlation	0.152	0.412 **	0.665 **	1
	Sign. (two-tailed)	0.131	0.000	0.000	
	*N*	100	100	100	100

*. The correlation is significant at the 0.05 (two-tailed) level. **. The correlation is significant at the 0.01 (two-tailed) level. PEQ: Postural Evaluation Questionnaire; BPE: Biomechanical Postural Evaluation.

**Table 4 ijerph-18-03507-t004:** Regression model.

Variables	Mod.1		Mod.2		Mod.3	
	R-square	Anova sign.	R-square	Anova sign.	R-square	Anova sign.
	0.262	0.0000	0.17	0.0000	0.271	0.0000
	B	Beta	B	Beta	B	Beta
(Constant)	1.292		3.02		1.183	
PEQ Score	0.237	0.512 **			0.197	0.426 **
BPE Score			0.59	0.412 **	0.184	0.129 *

* Sing. *p*-value 0.000. ** Sing. *p*-value 0.050. PEQ: Postural Evaluation Questionnaire; BPE: Biomechanical Postural Evaluation.

## Data Availability

The data presented in this study are available on request from the corresponding author. The data are not publicly available due to safety reasons connected to possible subtraction and alteration in publicly accessible repository.

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
