# Peer review of "Posture and Health: Are the Biomechanical Postural Evaluation and the Postural Evaluation Questionnaire Comparable to and Predictive of the Digitized Biometrics Examination?"

_ijerph, 2021, doi:10.3390/ijerph18073507_

Round 1
Reviewer 1 Report
Dear Authors:
The quality of the presentation of the presented results is poor and needs to be improved. Authors should consult the most appropriate mode of presentation for publication in a quality journal like this one.
They should consult the quality of presentation of the results and use as fundamental references for the discussion on the research topic:
Analyzing the use of accelerometers as a method of early diagnosis of alterations in balance in elderly people: A systematic review (doi: 10.3390/s19183883)
Validity and reliability of a tool for accelerometric assessment of static balance in women (doi: 10.1080/21679169.2017.1347707)
Percentiles and reference values for the accelerometric assessment of static balance in women aged 50–80 years (doi: 10.3390/s20030940)
After the thorough correction of the presentation of results and the review of the Discussion and its extension with the new references, the reevaluation of your manuscript will be taken into account.
Kind regards
Author Response
Response to Reviewer 1 Comments
Point 1: The quality of the presentation of the presented results is poor and needs to be improved. Authors should consult the most appropriate mode of presentation for publication in a quality journal like this one.
They should consult the quality of presentation of the results and use as fundamental references for the discussion on the research topic:
Analyzing the use of accelerometers as a method of early diagnosis of alterations in balance in elderly people: A systematic review (doi: 10.3390/s19183883)
Validity and reliability of a tool for accelerometric assessment of static balance in women (doi: 10.1080/21679169.2017.1347707)
Percentiles and reference values for the accelerometric assessment of static balance in women aged 50–80 years (doi: 10.3390/s20030940)
After the thorough correction of the presentation of results and the review of the Discussion and its extension with the new references, the reevaluation of your manuscript will be taken into account.
Kind regards
Response 1: Thank you very much for the Comment. The whole article has been revised, following the precious indications. We have changed the title, clarified the aim of the study, added literature references and improved the presentation of the results. We very much hope that the revised version will be found acceptable for publication.
Reviewer 2 Report
Authors solved all criticisms. Overall a good paper.
Author Response
Response to Reviewer 2 Comments
Point 1: Authors solved all criticisms. Overall a good paper.
Response 1: Thank you very much, we very much hope that the revised version will be found acceptable for publication.
Reviewer 3 Report
Title: This title does not reflect what this paper is trying to do. I would suggest stating something about comparing quesitionaries and BPE, to the the digital balance measure.
Abstract: The authors do not provide evidence that the myofascial is connected to emotional disorders.
Intro: The current does not present a reason why this study needs to be done. The authors explain spinal architecture but not why there is a need for low cost and easy to implement test of posture, which is the purpose stated in the paper. This needs to be focus of the intro.
Methods: Citations need to be provided for the BPE and PEQ. NO validation has been provided for these methods.
Table 1 may be missing some key features. It is unclear what this table is try to convey to the reader. No caption is provided.
Why was COP evaluated at 51.2 seconds? That seems quite random.
Results: Table 4 is very unclear. Why does it random change the B symbol to Beta?
Discussion: Lines 184 to 230 suffer from the same problem as the intro. It does not seem to relate to what the methods did or the purpose of the study. After these lines, the discussion does address this. Make this section clear to the readers.
Author Response
Response to Reviewer 3 Comments
Point 1: Title: This title does not reflect what this paper is trying to do. I would suggest stating something about comparing quesitionaries and BPE, to the the digital balance measure.
Response 1: Thank you very much for the Comment. We changed the title to "Posture and Health: are The Biomechanical Postural Evaluation and The Postural Evaluation Questionnaire comparable and predictive of the Digitized Biometrics Examination?". We hope it's a better title.
Point 2: Abstract: The authors do not provide evidence that the myofascial is connected to emotional disorders.
Response 2: We have added a bibliographic reference at Lines 245-252: “Indeed, basic muscle tone is the expression of multiple afferential informations from visceral, emotional and structural receptors. Each share at the spinal cord level is modulated by the autonomic nervous system (in particular by the orthosympathetic system), with influences on vascular function, internal organs and basic muscle tone. Therefore, in the vertical functions of the spinal cord there is a segmental (horizontal) manifestation, which expresses the dysfunction in the dermo-myotonic field. Therefore, myofascial and postural tonic dysfunction can be influenced by pre-existing pathologies or comorbidities, even asymptomatic ones [31].”
Point 3: Intro: The current does not present a reason why this study needs to be done. The authors explain spinal architecture but not why there is a need for low cost and easy to implement test of posture, which is the purpose stated in the paper. This needs to be focus of the intro.
Response 3: Lines 80-82: “A less expensive and quickly administrable postural evaluation could be administered in a preventive setting on a large scale, in order to prevent dysfunctional syndromes. This would have a positive impact on social costs and work activity.”
Point 4: Methods: Citations need to be provided for the BPE and PEQ. NO validation has been provided for these methods.
Response 4: We have mentioned the articles that analyze the relationship between postural tone alterations and myofascial, visceral and emotional disorders.
Point 5: Table 1 may be missing some key features. It is unclear what this table is try to convey to the reader. No caption is provided.
Response 5: We have updated the Table 1 by adding the caption " Table 1. Structures examined by the Biomechanical Postural Evaluation (BPE), divided by body districts.”
Point 6: Why was COP evaluated at 51.2 seconds? That seems quite random.
Response 6: Sorry for forgetting. We followed the AFP conditions. In this regard, we have added a bibliographic reference at Lines 129-131 [12].
Point 7: Results: Table 4 is very unclear. Why does it random change the B symbol to Beta?
Response 7: Sorry for misunderstanding. B symbol and Beta are two different coefficients. B is the regression coefficient, while Beta is the standardized regression coefficient. B is used to estimate the regression line for each model, while Beta is used to compare effects across models.
Point 8: Discussion: Lines 184 to 230 suffer from the same problem as the intro. It does not seem to relate to what the methods did or the purpose of the study. After these lines, the discussion does address this. Make this section clear to the readers.
Response 8: After Lines 219-221: “Other studies analyzed posture and balance strategies, but the samples always regard symptomatic individuals [25,26]. For instance, the balance in patients with chronic or post-surgical low back pain has been analyzed.”, we have added the following sentence: “This study, instead, is based on pain evoked and reported in asymptomatic individuals, in order to develop a protocol for the prevention of dysfunctional syndromes.” (Lines 221-223).
Reviewer 4 Report
The article improved a lot.
The writing is more fluent.
The paragraph of results sometimes seems like an "Italian English"
Line 220 I think the statement should me corrected.
The study design is well described however try to find out more articles about the validity of written tests or if it is more reliable than other kind of tests.
Author Response
Response to Reviewer 4 Comments
Point 1: The article improved a lot.
Response 1: Thank you very much, we paid close attention to the Comments. We very much hope that the revised version will be found acceptable for publication.
Point 2: The writing is more fluent.
Response 2: Thank you very much, we hope it is clearer.
Point 3: The paragraph of results sometimes seems like an "Italian English".
Response 3: We have corrected the paragraph of results. We hope that the English is of higher quality.
Point 4: Line 220 I think the statement should be corrected.
Response 4: We have corrected the statement at Lines 236-239: “Therefore, this study introduces an innovative, non-invasive, fast-performing, economical and repeatable method of investigation of posture, composing of a questionnaire and a biomechanics manual analysis. This method analyzes the posture researching and contextualizing every alteration which may influence the postural tone.”
Point 5: The study design is well described however try to find out more articles about the validity of written tests or if it is more reliable than other kind of tests.
Response 5: Thanks for the Comment. As suggested, we have added references 12 and 31.
“The stabilometric data were acquired on a force platform under standardized AFP conditions: subject barefoot, feet oriented at 30°, heels 2cm apart, arms alongside the body, visual target 90cm away in front of the subject, and acquisition time of 51.2 s [12]”.
“Indeed, basic muscle tone is the expression of multiple afferential informations from visceral, emotional and structural receptors. Each share at the spinal cord level is modulated by the autonomic nervous system (in particular by the orthosympathetic system), with influences on vascular function, internal organs and basic muscle tone. Therefore, in the vertical functions of the spinal cord there is a segmental (horizontal) manifestation, which expresses the dysfunction in the dermo-myotonic field. Therefore, myofascial and postural tonic dysfunction can be influenced by pre-existing pathologies or comorbidities, even asymptomatic ones [31].”
Round 2
Reviewer 1 Report
Dear Authors:
In the Introduction and / or Discussion they should also name other instrumental alternatives for biomechanical analysis such as accelerometers. They are highly relevant references in this field:
Percentiles and reference values for the accelerometric assessment of static balance in women aged 50–80 years. DOI: 10.3390 / s20030940
Validity and reliability of a tool for accelerometric assessment of static balance in women DOI: 10.1080 / 21679169.2017.1347707
Within-day test – retest reliability of an accelerometer-based method for registration of step time symmetry during stair descent after ACL reconstruction and in healthy subjects. DOI: 10.1080 / 09593985.2020.1723150
Personally, the table with the correlation matrix can be dropped and convey that information in the text.
A Conclusions section should be included with the final interpretation of the relevant and reliable findings of this research and its application and consequences for practice.
Kind regards.
Author Response
Response to Reviewer 1 Comment
Point 1: In the Introduction and / or Discussion they should also name other instrumental alternatives for biomechanical analysis such as accelerometers. They are highly relevant references in this field:
Percentiles and reference values for the accelerometric assessment of static balance in women aged 50–80 years. DOI: 10.3390 / s20030940
Validity and reliability of a tool for accelerometric assessment of static balance in women DOI: 10.1080 / 21679169.2017.1347707
Within-day test – retest reliability of an accelerometer-based method for registration of step time symmetry during stair descent after ACL reconstruction and in healthy subjects. DOI: 10.1080 / 09593985.2020.1723150
Response 1: Thank you very much for this comment. As suggested, we have mentioned the three articles in the text.
“DBE was considered in this study, but there are other tools that objectively evaluate static balance, such as triaxal accelerometers [11,12].” (Introduction: lines 80-11).
“This method, aimed at asymptomatic subjects, could also be applied to symptomatic subjects and of all ages. In fact, with increasing age there is a static balance deterioration [35], which could cause an increase in the risk of falls. Targeted intervention, based on specific exercises aimed at recovering skills related to postural control, could reduce the risk of falls and hospitalizations.” (Discussion: lines 260-263).
Point 2: Personally, the table with the correlation matrix can be dropped and convey that information in the text.
Response 2: Thanks for the comment, we have explained the table in the text. We hope the explication is clear.
Point 3: A Conclusions section should be included with the final interpretation of the relevant and reliable findings of this research and its application and consequences for practice.
Response 3: At lines 257-268, we have provided an interpretation of the results and ways of using the method in clinical practice.
“This study, on the other hand, suggests a protocol, based on a multidisciplinary approach, that can be carried out in any environmental situation, with a minimal economic impact and with a result that with its values can express the same values obtained by a DBE (R-square=0,271). This method, aimed at asymptomatic subjects, could also be applied to symptomatic subjects and of all ages. In fact, with increasing age there is a static balance deterioration [35], which could cause an increase in the risk of falls. Targeted intervention, based on specific exercises aimed at recovering skills related to postural control, could reduce the risk of falls and hospitalizations. This method also lays the foundations to undertake a path in the preventive and curative field on a large scale and on different subjects who can easily run into dysfunctional syndromes not only structural and myofascial, but also related to the whole body homeostasis. Specially in this pandemic period which makes people more sedentary, dysfunctional syndromes could be prevented and this would have a positive impact on social costs and work activity.”
Kind regards.
Reviewer 3 Report
None
Author Response
Thank you very much for approval.
Kind regards.
This manuscript is a resubmission of an earlier submission. The following is a list of the peer review reports and author responses from that submission.
Round 1
Reviewer 1 Report
Dear Authors:
This research does not fulfill many formal aspects of writing a scientific article.
In addition, it does not comply with the presentation rules of this magazine. Authors should thoroughly rewrite the article and then resubmit the manuscript for evaluation.
Kind regards.
Reviewer 2 Report
The authors report an interesting topic. Just few points:
- Lines 81-83 "100 "G.d’Annunzio" University healthy students.... average age is 23.38 years". what about co-pathologies about this students? Can co-pathologies affect the results?
- Lines 226-231. Please include a Conclusion Section at this point.
- In the Discussion Section (lines 173-176) more biomechanical cervical considerations should be taken into account. Please add these two very important references at this point: Montemurro N et al. The Y-shaped trabecular bone structure in the odontoid process of the axis: a CT scan study in 54 healthy subjects and biomechanical considerations. J Neurosurg Spine. 2019 Feb 1:1-8. doi: 10.3171/2018.9.SPINE18396. Patwardhan AG et al. Cervical sagittal balance: a biomechanical perspective can help clinical practice. Eur Spine J. 2018 Feb;27(Suppl 1):25-38. doi: 10.1007/s00586-017-5367-1.
- Does the paper present any limitations? please report them.
Overall a good paper.
Reviewer 3 Report
Overall, this manuscript did not sufficiently provide any scientific reasoning of why this study was conducted. The introduction did not provide a research question that was being studied, much less reasoning why the current methods were chosen. Without this foundation, it makes the results and discussion uninterpretable.
Reviewer 4 Report
Dear authors,
thanks for submitting your article.
I have some dubts.
Where the aim of the study is expressed?
I suggest you to write it in the introduction.
Also, the introduction is full of information but it doesn' t focus the aim of the study. I would have focused it on the disadvantages of the more common methods of postural analysis.
Furthermore there are some mistakes:
- Line 49 Would you mean "body system" when you write "apparatuses"
- Line 69 "because" is not folowed by any statements.
Anyway, please review the whole also because the paper is introduced by topics that do not match what is mentioned in the title (i.e. I expect to red about the novelty of new preventing postural imbalance methods).
The paragraph "materials and methods" could be improved.
You could arrange a table where the participants' features are descrived (nr. of people, age, sex, etc.....)
You should also report the inclusion and exclusion criteria.
The lines about the participants' privacy protection is well done.
However study design is not defined.
Is this a cross-sectional?
BPE and DBE should be descrived more.
PEQ is well descrived.
Line 204.What would you mean with "y" ?
In conclusion, I strongly suggest to underly the relevance of the study for the economical advantages it could pursue and not on the physiology behind the posture.
Morever I suggest to be more specifiic among the numerous postural imbalance that could be detected from your questionnaire.
It is easy to predict postural imbalance as long as the greatest part of the population is generally more sedentary than the past thus everyone could have al least one.
I suppose that there has been hard work behind this paper, the subject of the study could be very interesting, but it could be improved !